# Peri-Implant Bone Loss Measurement Using a Region-Based Convolutional Neural Network on Dental Periapical Radiographs

**DOI:** 10.3390/jcm10051009

**Published:** 2021-03-02

**Authors:** Jun-Young Cha, Hyung-In Yoon, In-Sung Yeo, Kyung-Hoe Huh, Jung-Suk Han

**Affiliations:** 1Department of Prosthodontics, School of Dentistry and Dental Research Institute, Seoul National University, Daehak-ro 101, Jongro-gu, Seoul 03080, Korea; starriet@snu.ac.kr (J.-Y.C.); drhiy226@snu.ac.kr (H.-I.Y.); pros53@snu.ac.kr (I.-S.Y.); 2Department of Oral and Maxillofacial Radiology, School of Dentistry and Dental Research Institute, Seoul National University, Daehak-ro 101, Jongro-gu, Seoul 03080, Korea

**Keywords:** peri-implant bone level, peri-implantitis, deep learning, convolutional neural network, machine learning, artificial intelligence, keypoint detection, radiographs

## Abstract

Determining the peri-implant marginal bone level on radiographs is challenging because the boundaries of the bones around implants are often unclear or the heights of the buccal and lingual bone levels are different. Therefore, a deep convolutional neural network (CNN) was evaluated for detecting the marginal bone level, top, and apex of implants on dental periapical radiographs. An automated assistant system was proposed for calculating the bone loss percentage and classifying the bone resorption severity. A modified region-based CNN (R-CNN) was trained using transfer learning based on Microsoft Common Objects in Context dataset. Overall, 708 periapical radiographic images were divided into training (*n* = 508), validation (*n* = 100), and test (*n* = 100) datasets. The training dataset was randomly enriched by data augmentation. For evaluation, average precision, average recall, and mean object keypoint similarity (OKS) were calculated, and the mean OKS values of the model and a dental clinician were compared. Using detected keypoints, radiographic bone loss was measured and classified. No statistically significant difference was found between the modified R-CNN model and dental clinician for detecting landmarks around dental implants. The modified R-CNN model can be utilized to measure the radiographic peri-implant bone loss ratio to assess the severity of peri-implantitis.

## 1. Introduction

In general, the determination of the peri-implant marginal bone level on the conventional radiographs is difficult since the three-dimensional bone shape needs to be comprehended from a two-dimensional image [1,2]. Oftentimes, the boundaries of the bones around the implant are not clear, or the heights of the buccal and lingual bone levels are different.

Meanwhile, deep learning methods for image recognition, such as convolutional neural networks (CNNs), were improved remarkably after AlexNet [3] won the ImageNet Large Scale Visual Recognition Challenge [4] in 2012. Since deep neural networks successfully classified general images, numerous types of methods have been developed and applied to medical images. Accordingly, a large number of studies have been conducted on CNNs for diagnosis using the binary classification of radiographic images, such as pulmonary tuberculosis [5], osteoporosis [6], or periodontal bone loss [7].

Recently, some branches of CNNs not only perform classifications but also predict the bounding boxes of objects. You Only Look Once (YOLO) [8] and Single Shot MultiBox Detector (SSD) [9] detect objects within a very short period of time. Moreover, Mask R-CNN, which is a modified architecture of the region-based convolutional neural network (R-CNN) [10], can localize bounding boxes around objects and also predict segmentation masks or keypoints within the boxes [11].

Compared to medical image classification, fewer studies have explored the detection of the locations of specific lesions. Some studies attempted to segment lesions on radiographs [12,13] or magnetic resonance imaging [14] based on semantic segmentation methods, such as U-Net [15]. However, there are few studies on instance segmentation with radiographs, for which each individual object needs to be detected. Moreover, research related to CNNs in the field of dentistry is relatively insufficient compared to the field of medicine [16]. Nevertheless, these machine learning approaches are currently one of the trends driving advances in dentistry, especially diagnostic imaging [17].

To the best of our knowledge, the detection of individual dental implants and location of important landmarks, such as an implant marginal bone level, using fully end-to-end deep learning methods has not yet been studied. The present study aims to address this lacuna in research. This study evaluated a deep learning model, namely, the Mask R-CNN, for localizing the implants and finding keypoints within the detected implant site. Based on the results, the marginal bone loss ratio was calculated, and the corresponding classification was performed. Such a classification may assist dentists in the analyses of periapical radiographs.

## 2. Materials and Methods

### 2.1. Datasets

For datasets, 1000 periapical radiographic images were obtained between December 2018 and June 2020 from the Picture Archiving and Communication System (PACS) in Seoul National University Dental Hospital. Among these, radiographs that were not properly taken in parallel or those that were out of focus were excluded. Moreover, if the graft material hindered the correct observation of the alveolar bone in a radiograph, it was excluded. Finally, a total of 292 periapical radiographs were excluded. The remaining 708 images were separated into upper and lower periapical radiographs. For both upper and lower datasets, the images were further classified into training, validation, and test datasets. The overview of the datasets is shown in Figure 1.

Training data were enriched by randomly generated data augmentation, leveraged with a horizontal flip, rotation, contrast, and brightness shift. The size of the augmented dataset was not fixed, as the data augmentation process used in this study could generate an infinite number of modified input images. However, subsequent to analyzing the evaluation result of the validation dataset, the training was stopped after 18,000 iterations to avoid overfitting, with the learning rates ranging from 0.0005 to 0.00005. Implant bounding boxes and keypoint annotations for ground truth were performed by a dental practitioner (J.-Y.C.), and an oral and maxillofacial radiologist (K.-H.H.) reviewed and confirmed the results.

### 2.2. Neural Network Model Architecture

For better results during the prediction of the landmarks around the implants, we prepared three separate neural networks and connected them. One is for identifying the upper or lower jaw. After the radiographic image was classified, it was fed into one of the two other networks, which are responsible for upper or lower implants, respectively. These two parallel networks proceed the core logic, detecting implants with bounding box regression and predicting landmarks. They have the same architectures but were separately trained for upper and lower implants.

#### 2.2.1. Upper and Lower Classifier

Based on a 152-layer deep residual learning neural network (ResNet) [18], a classification model was created for sorting upper and lower periapical radiographs. Weights pretrained on ImageNet [19] were used, but the last fully connected layer was switched so that the number of the final output nodes became two: the number of the classes (upper and lower implants in this study).

#### 2.2.2. Mask R-CNN with Keypoint Head

After the 152-layer model classified whether the given periapical radiograph contained an image of upper or lower jaw, the radiograph image was fed into another model that was trained specially for upper or lower implants.

In this phase, individual implants are detected with bounding boxes. Based on the detected region, the six keypoints, including mesial and distal marginal bone level, were predicted.

For this procedure, a modified R-CNN architecture was used, Mask R-CNN. Mask R-CNN, the latest descendant of the R-CNN model, comprised a “backbone” and “heads” [11]. The backbone network is a CNN that outputs feature maps from the original input image. It can be of various types, but the feature pyramid network (FPN) [20] based on ResNet [18] is known for robust results when used for Mask R-CNN, and, thus, it was adopted in this study.

Using the feature maps from the backbone network, the box head performed object classification and bounding box regression and the mask head performed the object segmentation task. By attaching a keypoint detection head and properly training the network, the model can predict specific keypoints on the objects that were detected by the box head. As shown in the previous study, this method with a keypoint head can be used for human pose estimation, wherein the model picks some keypoints of the human body, such as eyes, elbows, and knees [11]. In the present study, we adopted this architecture, the Mask R-CNN based on ResNet-FPN backbone with a keypoint detection head. The scheme of the model excluding the upper and lower classification is shown in Figure 2.

The model was trained and tested for detecting implants and locating the six positions of the detected implant on dental periapical radiographs. The six keypoints were peri-implant bone level, the implant apex, and the implant top, which all have right and left sides. To cover various types of implants, the most coronal thread was annotated as the top of the implant. We refer to these six positions as “keypoints” since it is a widely used terminology in point detecting tasks, such as human pose estimation [21,22] or facial keypoint detection [23].

#### 2.2.3. Bone Loss Ratio and Classification

Some studies exist on classification systems for peri-implantitis that use radiographic bone loss together with clinical indicators, such as bleeding/suppuration on probing or probing depth [24,25,26]. These studies used the ratio of the radiographic bone loss over the total implant length to classify the peri-implantitis. Based on the criteria suggested by these studies, we calculated and classified the bone loss ratio so that the dental practitioners can easily refer to it.

Using the coordinates of the six keypoints that resulted from the prediction, the total length of the implant and the implant length that are not surrounded by sound bone can be calculated. The total length was measured from the center of the apex to the center of the implant top, and the length corresponding to the radiographic bone loss was measured from the center of the implant top to the center of the two marginal bone level keypoints. From these values, the percentage of the implant length in the bone defect site over the total length was calculated.

Based on this percentage, the severity of the bone loss around the implant was classified. The severity was categorized into four groups: normal, if the percentage is ≤10%, early, if the percentage is >10% and ≤25%, moderate, if the percentage is >25% and ≤50%, and severe, if the percentage is >50%.

### 2.3. Evaluation Methods

Since the prediction process comprises two phases, which are implant bounding box regression and implant keypoint localization, two different metrics are used for evaluating each task.

#### 2.3.1. Intersection over Union (IoU)

For evaluating the model’s performance of detecting implants, a metric is needed that can measure how close the model’s bounding box is to the ground truth bounding box. To accomplish this, the Intersection over Union (IoU), also known as the Jaccard index, is used. IoU is calculated by dividing the overlapping area of the ground truth box (A) and the model-predicted box (B) by the total area of the coverage of the two boxes.
(1)$$\mathrm{IoU}=\frac{\left|A\cap B\right|}{\left|A\cup B\right|}$$

At various values of the IoU thresholds, the model’s average precision (AP) and average recall (AR) can be obtained.

#### 2.3.2. Object Keypoint Similarity (OKS)

To evaluate the model’s keypoint detection performance, object keypoint similarity (OKS) [27] is used as an analogous to IoU. OKS is calculated for each object, and it ranges from 0 to 1. The value tends closer to 1 as the model’s prediction gets closer to the ground truth. This metric can be used similarly to IoU, which is generally used for evaluating object detection tasks. The OKS per each implant is defined as follows.
(2)$${\mathrm{OKS}}_{j}=\frac{\sum_{i}\left[exp\left(\frac{-d_{ji}^{2}}{2s_{j}^{2}k_{i}^{2}}\right)\delta \left(v_{ji}>0\right)\right]}{\sum_{i}\delta \left(v_{ji}>0\right)}$$

In the above equation, j represents each individual implant and i corresponds to each keypoint type. In the present study, the i represents the Lt. bone level, Rt. bone level, Lt. apex, Rt. apex, Lt. implant top, and Rt. implant top. Furthermore, v are the visibility flags (v = 0: not labeled, v = 1: labeled but not visible, and v = 2: labeled and visible) of the ground truth. For each implant, the ground truth and predicted keypoints have the form [x_1_, y_1_, v_1_, …, x_6_, y_6_, v_6_], where x, y are the keypoint locations and v is a visibility flag.

Consider a vector ${\vec{d}}_{ji}$ that starts from a ground truth keypoint and ends at the detected keypoint, variable $d_{ji}$ represents the distance between the ground truth keypoint and the detected keypoint. Furthermore, $s_{j}$ is the scale of the implant j and is defined as the square root of the ground truth segmented area of the implant.

Moreover, $k_{i}$ can be regarded the per-keypoint standard deviation but multiplied by some constant, which is 2 herein$\;\left(k_{i}=2\sigma_{i}\right)$. To obtain the per-keypoint standard deviation $\sigma_{i}$, standardized to implant scale s, redundantly annotated images in the validation dataset were used to calculate $\sigma_{i}^{2}=E_{\left(j\right)}\left[d_{ji}^{2}/s_{j}^{2}\right]$. Here, $E_{\left(j\right)}$ represents an average over j. As the mean of $\frac{{\vec{d}}_{ji}}{s_{j}}$ over j becomes a zero vector, the per-keypoint standard deviation $\sigma_{i}$ can be obtained by calculating the mean of $d_{ji}^{2}/s_{j}^{2}$ over j.

OKS can be used as a threshold when determining precision and recall based on keypoint detection. Among the keypoint-detected implants, only the ones whose OKS value is higher than the OKS threshold are considered true positives. Using different settings of the OKS thresholds, the corresponding precision-recall curves as well as AP and AR can be obtained.

#### 2.3.3. Mean OKS

To compare the prediction result of the model with results of humans, all the OKS values of detected implants were averaged to calculate the mean OKS. While the precision-recall graphs reflect the model’s prediction confidence scores, the mean OKS does not include the information of various confidence scores. When comparing with a human, confidence scores cannot be used unless the human who performs the detection task provides specific confidence scores in a similar way to the model. Instead, only one threshold value of 0.7 was used for implant detection, and only the detections that had a confidence score above it were regarded valid when calculating the mean OKS. To verify the validity of the model, this metric was used to compare the model in this study to a dentist.

### 2.4. Keypoint Heatmap Visualization

The keypoint detection output of the deep learning model used in this study comprises six points. These points are determined by selecting the highest logit of the neural network’s output. By converting the logit values to colored keypoint heat maps, the likelihood of each pixel of being a keypoint can be visualized. Hence, the pixels that were given high scores by the model become easier to find. The keypoint logits were converted to values between 0.0 and 1.0, which were consequently assigned to a specific color. Examples of the keypoint heat maps are shown in Figure 3.

### 2.5. Statistical Analysis

Varying the threshold for the model’s confidence scores on bounding box regression with different IoU thresholds, the AP and AR on implant detection were obtained (IoU threshold 0.50–0.95, increased by 0.05). In addition, using different thresholds for the model’s confidence scores on keypoint detection with different OKS thresholds, the corresponding precision–recall curves and the AP and AR were obtained (OKS threshold of 0.50–0.95, and increased by 0.05). An independent *t*-test was used to compare the mean OKS values between a dentist and our model with total, upper, and lower dataset. *p*-value < 0.05 was considered to be statically significant. The calculation of the AP, AR, precision–recall curves, and the Independent *t*-test results was processed with Python (Python 3.6.9, Python Software Foundation). The software code used for the evaluation was modified from an open-source project of previous research [28].

## 3. Results

### 3.1. Implant Detection Evaluation

For evaluation of implant detection (bounding boxes around implants), AP and AR at various IoU thresholds were calculated. The AP and AR for the upper implant detection averaged for all IoU thresholds increased by 0.05 from 0.5 to 0.95 are 0.627 and 0.684, respectively. The AP and AR for the lower implant detection averaged for all IoU thresholds are 0.657 and 0.728, respectively. The results are presented in Table 1.

### 3.2. Keypoint Detection Evaluation

To obtain the OKS value in Equation (2), the standard deviations $\sigma_{i}$ for each keypoint type i around the implant were calculated by annotating the test dataset twice. The calculated $\sigma_{i}$ for this study were as follows: $\sigma_{i}$ = [0.0895, 0.0816, 0.0193, 0.0196, 0.0209, and 0.0273] for Lt. bone level, Rt. bone level, Lt. apex, Rt. apex, Lt. implant top, and Rt. implant top, respectively. Based on these standard deviations, the OKS values for each ground truth and prediction pair were calculated. Similar to the IoU, OKS value can be interpreted as how close the model’s prediction is to the ground truth keypoints.

Using these results, AP and AR at various OKS thresholds were calculated for the evaluation of keypoint detection around the detected implant. The results are described in Table 1. In addition, Precision–Recall curves were crafted by varying the OKS threshold from 0.50 to 0.95 in increments of 0.05. The graphs are shown in Figure 4.

### 3.3. Mean OKS

For the test dataset, the mean OKS values of the model used in this study were 0.8748, 0.9029, and 0.8885 for upper, lower, and total test datasets, respectively, while the mean OKS of a dentist for the total test dataset was 0.9012. From Equation (2), the individual keypoint similarity is $exp\left(-d_{i}^{2}/2s^{2}k_{i}{}^{2}\right)$ and $k_{i}=2\sigma_{i}$. Thus, given the mean OKS, where the prediction belongs within the normal distribution of human annotated keypoints can be estimated. The mean OKS of 0.8885 for the total test dataset corresponds to $d_{i}/s$
≈
$0.9725\sigma_{i}$, on average, and this indicates that approximately 66.92% of human keypoint annotations are better than that of the model used herein. To compare the mean OKS values between a dentist and our model, an independent t-test was performed. All pairs showed no statistically significant difference. The results are shown in Table 2.

## 4. Discussion

A dental clinician needs to locate appropriate landmarks to analyze radiographs and diagnose peri-implantitis. Moreover, as suggested by previous studies, the severity of peri-implantitis can be categorized based on the percentage of the radiographic bone loss [24,25,26]. For these tasks, the automated system proposed herein can be used for assisting dental researchers or practitioners. The examples of the predicted results are shown in Figure 5.

AP and AR are widely used evaluation metrics in the field of object detection and instance segmentation. However, these metrics are intended for model evaluation and not human evaluation because AP and AR are calculated using scores, indicating the model’s confidence. Thus, the application of these metrics for evaluating humans is difficult. For a long time, the prediction results of machines were far from those of humans with regard to object detection or instance segmentation [4]. Therefore, the metrics comparing AI and human in these fields have not been extensively studied.

Meanwhile, in the field of segmenting biological images, such as cell segmentation, the average IoU [29,30,31], or the mean Dice coefficient [32,33] are frequently used. These metrics can also be applied to humans because they do not require a confidence score. As OKS was designed analogous to IoU, OKS can be averaged and interpreted similarly to the average IoU. Herein, to compare the prediction results of the model with those of a human, the mean OKS was calculated over entire implants that were detected in the test dataset. The result of the independent *t*-test showed no statistically significant difference between a dentist and the model. Considering the *p*-values, this deep learning model was considered to be helpful in detecting peri-implantitis under the clinical situation.

To measure the amount of the bone loss, a reference position should be present that can be considered as a threshold for a sound bone level. According to the 7th and 8th European Workshop on Periodontology [34,35], the use of a baseline radiograph is recommended after physiologic remodeling, which is usually at the time of prosthesis installation unless immediate loading is performed, to assess the changes in the level of the crestal bone. However, in many clinical situations, a baseline radiograph is unavailable. Such cases were discussed in previous studies. The 8th European Workshop on Periodontology reached a consensus that a vertical distance of 2 mm from the expected marginal bone level is recommended for a threshold when no previous radiograph is present [34]. Additionally, in the 2017 World Workshop on the Classification of Periodontal and Peri-Implant Diseases and Conditions, the threshold was set to 3-mm apical of the most coronal portion of the intra-osseous part of the implant in the absence of a previous radiograph [36]. Other studies suggested a threshold from a fixed reference point, such as 2 mm apically from the implant platform for bone-level implants or 2 mm apically from the apical termination of the polished collar for tissue-level implants [24,37].

Since we did not use baseline radiographs, as is the usual case in many clinical situations, setting a reference point that can be used for a bone loss threshold is important. Due to the fact that finding the expected marginal bone level can be subjective and the 2 mm distance in the radiographs may vary owing to distortion or magnification, a clear landmark is required that can be identified on radiographs.

Our dataset included a wide variety of implants, which have different shapes and implant-abutment junctions. If the implant is a bone-level implant and adopted platform switching, the most coronal position of the implant can be clearly observed. However, some types of implants, such as tissue-level implants, have shapes for which identifying the most coronal point of the rough surface on periapical radiographs is difficult.

To cover all the implants in the dataset and those used widely in clinical practice, the most coronal thread of the implant was adopted as a threshold position. In other words, the most coronal thread was considered as a reference point that is supposed to be a sound bone level if there was no bone resorption of more than 0.2 mm annually after the physiologic remodeling, which occurs mostly during the first year after implant placement. This can avoid the radiographic distortion or magnification that will make it difficult to determine a 2-mm distance in the radiographs. Furthermore, this approach can be applied to various types of implants. However, this method has a limitation since some types of implants have the most coronal thread at a more apical position than other types of implants. For instance, the reference point of the implant top for tissue-level implants should be around 2 mm below the apical termination of the polished collar, but, in many cases, the most coronal thread is located below that. Nevertheless, identifying the point that is the end of the rough surface area or the apical termination of the polished collar on periapical radiographs even for a human is difficult. Further research is necessary to overcome this problem.

Numerous previous studies have mentioned diagnosis criteria for peri-implantitis [24,34,35,36,37]. Most of them use radiographic marginal bone loss and bleeding on probing and/or suppuration as the criteria. As the diagnosis of peri-implantitis requires both radiographic and clinical information, diagnosing the disease only with periapical radiographs is not possible. This is a limitation of the suggested system herein, and further research needs to be conducted using more general information, including clinical information to assist peri-implantitis diagnosis. In addition, some information such as the length of the implant is needed to obtain the absolute depth of the bone defect site. With the ratio calculated by the automated system suggested in this study, the absolute bone loss length can be obtained by multiplying the real length of the implant.

It is also important to note that two-dimensional images are not the only tool to evaluate the marginal bone loss of the dental implant. Cone-beam computed tomography (CBCT) is helpful when assessing the peri-implant bone loss since it can provide a three-dimensional relationship between a dental implant and the surrounding alveolar bone. Some previous studies have sought to identify bone conditions around implants using periapical radiographs and CBCT images together [38,39], and other studies have reported that CBCT has a high accuracy for detecting peri-implant bone defects [40,41]. Thus, it will be more meaningful if a machine learning system can utilize the information from the CBCT as well as two-dimensional radiographs when evaluating the peri-implant bone conditions. This should be further studied in the future. Still, measuring the amount of the bone defect on conventional radiographic methods is important since two-dimensional radiographic images are widely used in the field and dental clinicians often encounter situations in which CBCT cannot be used due to the financial reasons of patients or the circumstances of dental clinics.

Although we can compare the mean OKS between the model and a human, the method has some limitations. First, a threshold for the model still needs to be set, over which the model will output the bounding box results. Hence, object detection results and the mean OKS value can vary with the threshold. In addition, as the information of the model’s confidence score above the threshold is not used, the comparison is limited between two models whose confidence scores are different.

Second, the standard deviations *σ_i_*, for each keypoint type *i* around the implant was calculated based on human (i.e., dentist) annotations. However, even for a dentist, bone level detection is a challenge, as seen from the following values: *σ* = [0.0895, 0.0816, 0.0193, 0.0196, 0.0209, 0.0273]. The first two figures show the standard deviations of the left and right bone level annotations, which are the greatest among the six values.

Thus, if the model precisely locates the bone level and locates other keypoints far from the ground truth, the OKS value decreases even though the model was successful on more challenging tasks. Since the standard deviations *σ_i_* of human annotations are larger for bone level keypoints, the distances of the detected bone level keypoints from the ground truth have relatively low contribution. Since we consider localizing the bone level as the most important task, this evaluation metric should be modified in further research.

Unlike the common objects in a context (COCO) evaluation method [27] where the object detection score of the model is used for applying a threshold when plotting the precision–recall curves, the keypoint localizing score was used herein. For tasks where the difficulty of detecting objects is proportional to that of localizing keypoints, using the object detection score for the precision–recall curve makes sense. However, in our task, detecting the implant was easier than detecting the keypoint position and the confidence score of implant detection was very high for many cases. This implies that the confidence score of implant detection does not indicate the confidence rate of keypoint detection. The plotted results of the precision–recall curves using the two different confidence scores were different in our task. When using the implant detection score, even when the recall value decreased, the precision did not reach close to 1.0 when the OKS threshold was high (>0.70). After changing the confidence score to represent the keypoint score instead of the implant detection score, the precision reached 1.0 at all OKS thresholds as the confidence score threshold increases. However, the results may vary because of the randomness of the data augmentation process during training.

## 5. Conclusions

After fine-tuning with a machine learning method based on transfer learning, the ability of the Mask R-CNN model to determine the extent of bone loss on radiographs for diagnosing peri-implantitis has reached the level of experts. Thus, the automated method based on this model can be used to assist dentists in diagnosing and categorizing peri-implantitis.

## Figures and Tables

**Figure 1 jcm-10-01009-f001:**
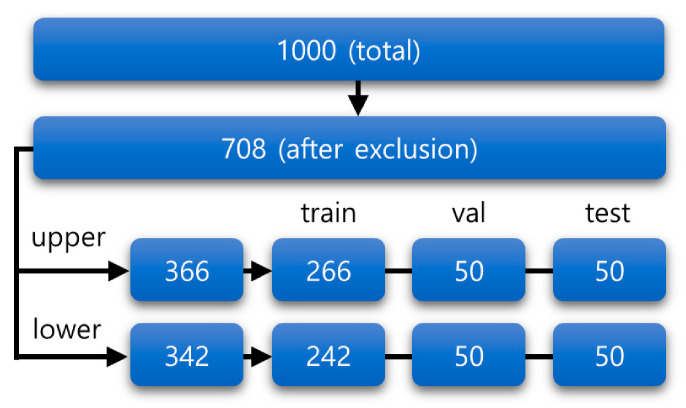
Overview of the datasets. After excluding the data with the exclusion criteria, remaining data were separated into train, validation, and test datasets. The train dataset was used for training the model, and the validation dataset was used for assessing the overfitting. The test dataset was used for evaluation. Each digit represents the number of periapical radiographs. The upper and lower periapical radiographs are shown separately.

**Figure 2 jcm-10-01009-f002:**
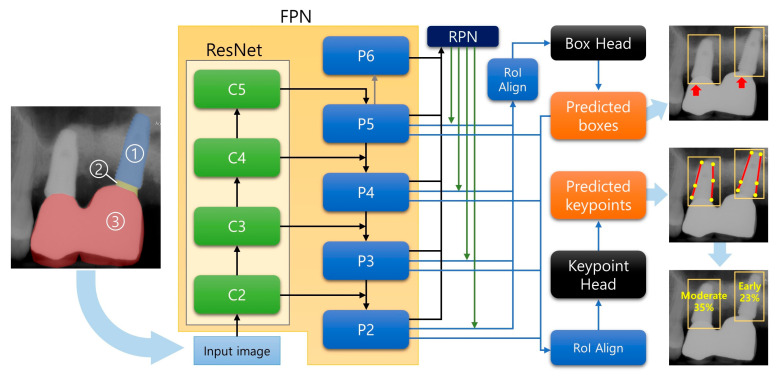
Architecture of the model used in this study. The region proposal network (RPN) takes feature maps from the feature pyramid network (FPN) and proposes region of interest (RoI). The box head further refines the proposals and predicts final bounding boxes (red arrows). In addition, the keypoint head localizes the keypoints (yellow dots shown in the middle of the radiographs on the right) based on the predicted bounding boxes. C2-5 and P2-6 denotes the output feature maps of residual network (ResNet) and FPN, respectively. The upper and lower classification phase is not shown for brevity. Implant (①), Abutment (②) and Superstructure (③) are shown in the left radiograph. Abbreviations: FPN, Feature Pyramid Network; ResNet, Residual Network; RPN, Region Proposal Network; RoI, Region of Interest.

**Figure 3 jcm-10-01009-f003:**
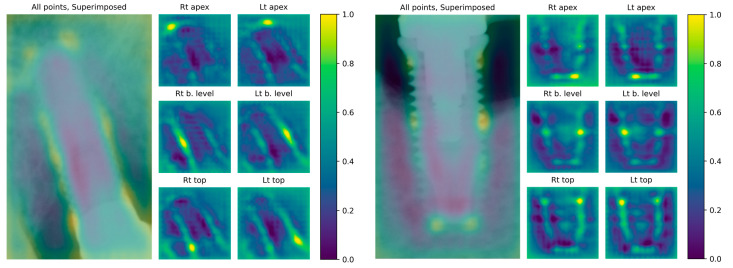
Examples of the keypoint heat maps. The likelihood of each pixel of being a keypoint is mapped into the heat map. Heat maps that combine all keypoints are superimposed on the original radiograph, and individual heat maps for each corresponding keypoint are shown.

**Figure 4 jcm-10-01009-f004:**
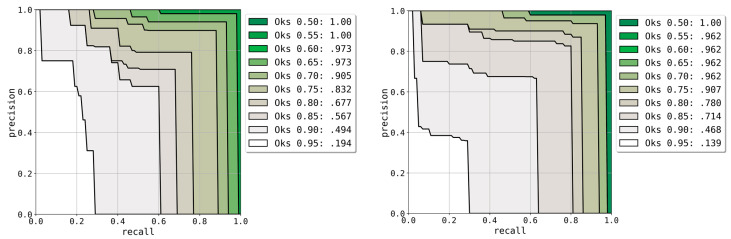
Precision–Recall graph for various object keypoint similarity (OKS) thresholds. Each colored graph represents each OKS threshold, and each point in the graph corresponds to a specific confidence score threshold of the model. Left: result of the upper images. Right: result of the lower images.

**Figure 5 jcm-10-01009-f005:**
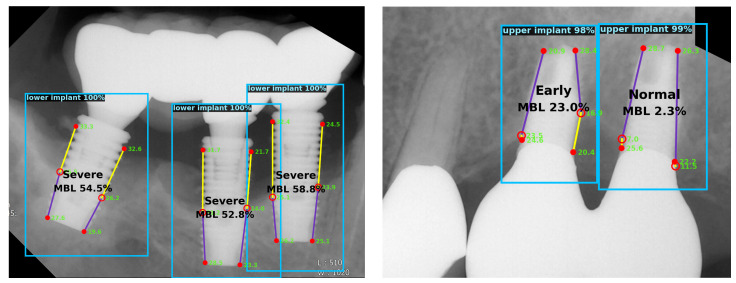
Examples of the predicted results. Each implant is detected with a bounding box and predicted keypoints are shown within the box. Radiographic bone loss ratio is calculated based on the keypoint locations. Confidence scores of the implant and keypoint detection are also shown.

**Table 1 jcm-10-01009-t001:** Average precision (AP) and average recall (AR) on various intersections over union (IoU) and object keypoint similarity (OKS) thresholds.

		AP (All)	AP (50)	AP (75)	AR (All)
bounding box	upper	0.627	1.000	0.746	0.684
	lower	0.657	1.000	0.714	0.728
keypoints	upper	0.761	1.000	0.832	0.810
	lower	0.786	1.000	0.907	0.845

AP (all): AP averaged over all IoU/OKS (0.50–0.95). AP(50): AP at IoU/OKS 0.50. AP(75): AP at IoU/OKS 0.75. AR (all). AR averaged over all IoU/OKS (0.50–0.95).

**Table 2 jcm-10-01009-t002:** Mean object keypoint similarity (OKS) values of a dentist and the model.

	Mean OKS		*p*-Value
Dentist	0.9012	Dentist–Model (total)	0.4095
Model (total)	0.8885	Dentist–Model (upper)	0.1441
Model (upper)	0.8748	Dentist–Model (lower)	0.9125
Model (lower)	0.9029	Model (upper)–Model (lower)	0.1543

## Data Availability

Restrictions apply to the availability of these data. Data used in this study was obtained from Seoul National University Dental Hospital and are available with the permission of Institutional Review Board of Seoul National University Dental Hospital, Dental Life Science Research Institute.

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
