# Peer review of "Peri-Implant Bone Loss Measurement Using a Region-Based Convolutional Neural Network on Dental Periapical Radiographs"

_jcm, 2021, doi:10.3390/jcm10051009_

Round 1

Reviewer 1 Report

Dear authors !

I was given the opportunity to review Your interesting paper investigating AI-based determination of bone loss around dental implants based on common intra-oral dental radiographs.

The entire manuscript is well-structured, written in correct English language and methodology documented comprehensibly. The figures aid the comprehension of the experimental setup. References are up to date and discussion highlights the novelty in periimplant diagnosis, it´s unbiased reliability but also the limitations.

The application of neural network based AI in dental radiograph interpretation and diagnosis is a novelty.

It was a pleasure to review this manuscript.

Author Response

Dear reviewer,

We greatly appreciate your very positive comments. 
It is an honor to be able to show this manuscript to a special reviewer because it is not easy to comprehensively understand this study as it covers two different fields. 
As you may well know, recent researches on applying machine learning, especially deep learning to medicine and dentistry are being actively conducted. We hope that our research will motivate more researchers in related fields. Thank you.

With best regards,

Jun-young Cha, Hyung-In Yoon, In-Sung Yeo, Kyung-Hoe Huh, and Jung-Suk Han

Reviewer 2 Report

The manuscript presents a interesting new content. However some major issues needs to be resolved.

“In general, the determination of the peri-implant marginal bone level on the radiographs is difficult since the three-dimensional bone shape needs to be comprehended32from a two-dimensional image”

Cited reference are form 2013 and 2007.

It is not true that only 2D images can be used in assessment of the marginal bone loss. CBCT image should be considered since you can evaluate the 4 points around the implant. There are several RCT that use CBCT 3D images to calcualte the buccal bone loss since it is not visible on 2D periapical x-ray.

Low emission of X-rays should encourage the use of modern X-ray point techniques such as CBCT

Material and Methods

The methodology is interesting, but in my opinion, for credibility, the study should be planned differently. Analysis of the 2D and 3D image of the same patient, the same implant, in this situation machine learning was used to “teach” the computer to recognize features in 2D images that indicate, for example, buccal bone loss.

Here again it is not entirely true, since even the complete buccal or palatal bone loss will not be observed on the 2D periapical X-ray.

Moreover in the clinical for the assessment of the bone loss is given in millimeters not in percentages depending of implant length (line 157-160 of the presented manuscript). Bone loss should not be assessed in terms of implant length, but in absolute units. This is a fatal error in the proposed methodology. To present the values in mm it requires calibration of the 2D radiological image. For example, knowing the length of the implant, one can calibrate a 2D periapical image to determine the MBL bone loss in millimeters.

There are different types of implants and different types of abutments. It seems that the proposed method is only applicable to implants that are platform switching. Therefore, it cannot be used for the entire group of implants without platformswitching (ex. Thommen Medical), tissue level implants (eg Straumann Tissue level), etc.

It is difficult for me to refer to IT methodology as it is a current issue for me. However, the developed method has no clinical value in its current form, nor does it constitute a parameter evaluation tool for the purposes of scientific work.

Limitations of the study should be discussed.  

Round 2

Reviewer 2 Report

The changes requested were made and answers are comprehensive. The article has improved, but some minor issues still need to be clarified:

In the discussion, the issue of CBCT and MBL measurement should be discussed in the aspect of the latest publications in this field, both experimental studies:  

Song D, Shujaat S, de Faria Vasconcelos K, Huang Y, Politis C, Lambrichts I, Jacobs R. Diagnostic accuracy of CBCT versus intraoral imaging for assessment of peri-implant bone defects. BMC Med Imaging. 2021 Feb 10;21(1):23. doi: 10.1186/s12880-021-00557-9. PMID: 33568085; PMCID: PMC7877020.

And practical application of CBCT in clinical trials:

Hadzik J, Krawiec M, Kubasiewicz-Ross P, Prylińska-Czyżewska A, Gedrange T, Dominiak M. Short Implants and Conventional Implants in The Residual Maxillary Alveolar Ridge: A 36-Month Follow-Up Observation. Med Sci Monit. 2018 Aug 14;24:5645-5652. doi: 10.12659/MSM.910404. PMID: 30104560; PMCID: PMC6104555.

 and clinical use of the CBCT combined with periapical x-ray to measure MBL

 Hadzik J, Krawiec M, SÅ‚awecki K, Kunert-Keil C, Dominiak M, Gedrange T. The Influence of the Crown-Implant Ratio on the Crestal Bone Level and Implant Secondary Stability: 36-Month Clinical Study. Biomed Res Int. 2018 May 16;2018:4246874. doi: 10.1155/2018/4246874. PMID: 29862269; PMCID: PMC5976988.

 Regarding measurement of bone defects (giving absolute values in millimeters) in the aspect of periimplantitis, it is worth referring to Workshop 2017, where the occurrence of periimplantitis is dependent on progressive/or up to 2mm MBL and symptoms of inflammation in the form of BOP.

Berglundh T, Armitage G, Araujo MG, Avila-Ortiz G, Blanco J, Camargo PM, Chen S, Cochran D, Derks J, Figuero E, Hämmerle CHF, Heitz-Mayfield LJA, Huynh-Ba G, Iacono V, Koo KT, Lambert F, McCauley L, Quirynen M, Renvert S, Salvi GE, Schwarz F, Tarnow D, Tomasi C, Wang HL, Zitzmann N. Peri-implant diseases and conditions: Consensus report of workgroup 4 of the 2017 World Workshop on the Classification of Periodontal and Peri-Implant Diseases and Conditions. J Clin Periodontol. 2018 Jun;45 Suppl 20:S286-S291. doi: 10.1111/jcpe.12957. PMID: 29926491.
